# Vitamin D-mediated effects on airway innate immunity *in vitro*

**Emma M. Stapleton**◉⊘, **Kathy Keck**⊘, **Robert Windisch**⊘, **Mallory R. Stroik**⊘, **Andrew L. Thurman**⊘, **Joseph Zabner**⊘, **Ian M. Thornell**◉⊘, **Alejandro A. Pezzulo**⊘, **Julia Klesney-Tait**⊘, **Alejandro P. Comellas**◉*⊘

Department of Internal Medicine, Roy J. and Lucille A. Carver College of Medicine, University of Iowa, Iowa City, IA, United States of America

⊘ These authors contributed equally to this work.
* Alejandro-comellas@uiowa.edu

## Abstract

### Introduction

Vitamin D supplementation has been suggested to enhance immunity during respiratory infection season. We tested the effect of active vitamin D (calcitriol) supplementation on key airway innate immune mechanisms *in vitro*.

### Methods

Primary human airway epithelial cells (hAECs) grown at the air liquid interface were supplemented with $10^{-7}$ M calcitriol for 24 hours (or a time course) and their antimicrobial airway surface liquid (ASL) was tested for pH, viscosity, and antibacterial and antiviral properties. We also tested hAEC ciliary beat frequency (CBF). Next, we assessed alterations to hAEC gene expression using RNA sequencing, and based on results, we measured neutrophil migration across hAECs.

### Results

Calcitriol supplementation enhanced ASL bacterial killing of *Staphylococcus aureus* ($p = 0.02$) but did not enhance its antiviral activity against 229E-CoV. It had no effect on ASL pH or viscosity at three timepoints. Lastly, it did not affect hAEC CBF or neutrophil migration, although there was a trend of enhanced migration in the presence of a neutrophil chemokine ($p = 0.09$). Supplementation significantly altered hAEC gene expression, primarily of AMP-related genes including *CAMP* and *TREM1*.

### Conclusion

While vitamin D supplementation did not have effects on many airway innate immune mechanisms, it may provide a useful tool to resolve respiratory bacterial infections.

**Data Availability Statement:** All relevant data are within the manuscript and its Supporting Information files.

**Funding:** This study was carried out with funding support from 5P01HL091842-13 (JZ): https://

grantome.com/grant/NIH/P01-HL091842-13-5864.
The funders had no role in study design, data
collection and analysis, decision to publish, or
preparation of the manuscript.

**Competing interests:** The authors have declared
that no competing interests exist.

## Introduction

Vitamin D is a well-established immunomodulator and potentiator of airway innate defense. Two observations support that vitamin D is associated with a reduction in respiratory infection risk. First, those with greater serum vitamin D $(25(OH)D_3)$ concentrations have a lower risk of respiratory tract infections and demonstrate higher lung function [1, 2]. Second, respiratory infection incidence increases during winter, when serum vitamin D levels are at their nadir [3, 4]. Vitamin D deficiency is also considered to be a risk factor for pulmonary inflammatory diseases [5], and respiratory exacerbations (asthma, chronic obstructive pulmonary disease, chronic rhinosinusitis with nasal polyps) [6–8]. Lastly, a recent meta-analysis of randomized controlled clinical vitamin D supplementation trials demonstrated that daily or weekly vitamin D supplementation provided a significant protective effect against acute respiratory tract infection, with stronger protective effects in participants with lower baseline serum vitamin D levels [9].

Calcitriol is a synthetic vitamin D analog that binds to the vitamin D receptor to increase serum blood calcium [10]. It can transport calcium and phosphate ions across epithelial cells [11]. Human airway epithelial cells (hAECs) constitutively generate calcitriol [12], and vitamin D increases antimicrobial and anti-inflammatory airway epithelial cell response *in vitro* [13, 14].

Airway innate immune defense consists of a variety of independent and coordinated mechanisms. First, antimicrobial peptides (AMPS), whose potency is affected by pH regulation, are embedded within airway surface liquid (ASL) and can immediately kill incoming viral and bacterial pathogens [15], After this first line of ASL defense, surviving pathogens can be cleared by mucociliary transport, which propels pathogens toward the larynx via ciliary beating [16]. Lastly, neutrophils phagocytose residual debris [17, 18]; Neutrophils express Triggering Receptor Expressed on Myeloid Cells 1 (*TREM1*) which can augment the immune response to infection and trigger an inflammatory cascade [18]. The role of calcitriol supplementation on ciliary beat frequency (CBF) or neutrophil migration in hAECs remains undetermined.

We hypothesized that calcitriol supplementation of hAEC media would improve ASL antiviral and antibacterial properties. We tested this *in vitro* by viral (229E-CoV) and bacterial (*Staphylococcus aureus*) challenge. Secondarily, we hypothesized that calcitriol supplementation would affect requisite airway host defense mechanisms, including ASL pH regulation, ASL viscosity, hAEC CBF and genetic expression. Lastly, we assessed whether increased *TREM1* expression after calcitriol supplementation would lead to increased neutrophil migration across hAECs.

## Methods

Previous viral and immunological research in human airway cells employed a calcitriol supplementation dose of $10^{-7}$ M in airway media [19–22]. Therefore, we applied a $10^{-7}$ M calcitriol (Tocris Bioscience, Bristol, UK) dose to basolateral media of primary bronchial/tracheal hAECs cultured at the air liquid interface (ALI) [23, 24] maintained with 5% $CO_2$ at 37˚C. Briefly, human donor lungs obtained following informed written consent were provided by the Iowa Donor Network, then dissected and seeded onto collagen coated filters at the ALI by the University of Iowa Cell Culture Core Repository. The study is approved by the University of Iowa Institutional Review Board (IRB #199507432). hAEC basolateral media consisted of DMEM/F-12 with 1% penicillin-streptomycin, 50 mg/ml gentamicin, and 2% Ultroser G (USG, Pall BioSepra, Cergy, France). The exact composition of USG is confidential, but correspondence with the manufacturer indicates it does not contain calcitriol. After 22-24h (unless otherwise noted), the following experiments and analyses were performed on donor-paired

samples. GraphPad Prism Software (version 9.0.2) was used for statistical analysis, and data are presented as individual data points or mean ± SEM. Significance was determined at $p < 0.05$.

## ASL antimicrobial activity against *S. aureus*

Bioluminescent *S. aureus* (Xen29, Caliper Lifesciences Bioware®) sub-cultured to log-phase growth was suspended in minimal media (5% TSB) and challenged (1:1) by ASL from calcitriol supplemented hAECs and by un-supplemented hAEC ASL [25]. Briefly, ASL was collected by serially washing (20 μl/well DI $H_2O$) hAEC apical surfaces from paired donor cells (n = 4 per experiment), after which media was changed back to baseline media for a >72h washout period. Subsequent calcitriol treatments were always applied to previously supplemented cells. Relative light units (RLUs), previously correlated to colony forming units [26], were read and immediate killing quantified by comparing RLUs within 12 minutes to RLUs at 0 min. To assess whether ASL was bactericidal, summary data (% live bacteria) per condition were compared to a hypothetical mean of 100 using a one-sample t-test. To assess differences between conditions, significance was assessed by paired *t*-test. All studies were performed in triplicate or quadruplicate. Vehicle and bleach controls were used to confirm ASL bacterial killing (**S1 Fig**). We also ensured that *S. aureus* growth between untreated and calcitriol-treated media was not different (**S2 Fig**).

## ASL antiviral activity against 229E-CoV

Donor-paired (n = 4/experiment) ASL from supplemented and un-supplemented hAECs or vehicle control was combined with human coronavirus 229E-CoV (ATCC® VR-740™) for 10 min at 33˚C, then applied to MRC-5 cells (ATCC® CCL171™) for 1h (33˚C + 5% $CO_2$). After 4–7 days we assessed plaque formation, as previously described [27]. Statistical significance was assessed by paired *t*-test and all studies were performed in triplicate.

## Effect of calcitriol supplementation on hAEC ASL pH

To assess the effect of calcitriol supplementation on ASL pH, apical hAEC surfaces were treated with 40 μl warm saline +1 mM HEPES solution (Research Products International, Mt Prospect, IL, USA) concurrently with basolateral media changes. After 22h, apical fluid was collected, and pH read by a pH microelectrode (Orion 9810BN, Thermo Scientific). Duplicate or triplicate paired samples from 9 donors were collected and significance determined by paired *t*-test.

## Effect of calcitriol supplementation on ASL relative viscosity

FITC-conjugated to 70kDa dextran (D1822; Thermo Fisher Scientific, Waltham, MA, USA) was applied into ASL using a sterile 100 μm nylon strainer (352360; Falcon, Durham, NC, USA) and left to absorb for ≥12 hours. For photobleaching experiments, cells were placed in fresh cell culture media ± calcitriol within a 37˚C chamber with 5% $CO_2$. FITC was excited with a 488 nm laser then two baseline images were acquired (Zeiss LSM880 microscope). Then, a small circular region (radius 5.2 μm) was photobleached and fluorescent recovery monitored for >1 min. Fluorescent recovery within ASL was compared to experiments with the FITC-dextran in phospho-buffered saline. ASL viscosity is represented as the relative viscosity to the saline sample.

## Effect of calcitriol supplementation on hAEC ciliary beat frequency

To test the effect of calcitriol on CBF, we applied supplemented media to the basolateral surface of hAECs (n = 4 and n = 8, depending on timepoint) after first measuring baseline CBF. CBF was acquired using brightfield imaging (Zeiss LSM880) across timepoints (45 min, 90 min, 24h). Line scanning mode was used to acquire images every 1.02 milliseconds (~980 Hz). A 40x LD C-Apochromat 40x/1.1 W Korr M27 objective (Carl Zeiss) was used to obtain images. hAECs were studied within a 37˚C chamber containing 5% $CO_2$. Within a single image, line scans often captured several ciliated cells. Each line scan was averaged into one value and five line-scans were performed per donor. We confirmed results by obtaining microscopic images of CBF using a Zeiss (Carl Zeiss AG, Oberkochen, Germany) Axio Observer microscope, studied under 37˚C and ambient air.

## Effect of calcitriol supplementation on hAEC gene expression

We tested the effect of calcitriol supplementation on hAEC gene expression using RNA sequencing. After 24h of treatment, mRNA was isolated from paired hAEC (n = 6) on polycarbonate filters, using the RNeasy Lipid Tissue Kit (Qiagen) according to the manufacturer's instructions. Genomic DNA was digested using DNase I (Qiagen). RNA samples were then quantified using fluorimetry (Qubit 2.0 fluorometer; Life Technologies), and RNA quality was assessed using an Agilent BioAnalyzer 2100 (Agilent Technologies).

## RNA library prep and sequencing

Library preparation and sequencing were conducted at the Iowa Institute of Human Genetics Genomics Sequencing Division core facility, University of Iowa. Briefly, an Illumina Stranded mRNA Prep kit was used to isolate oligo-dT purified polyadenylated RNA. The samples were then reverse transcribed to create cDNA. The cDNA was fragmented, blunt-ended, and ligated to indexed adaptors. Following quantification of the cDNA generated for the library, the samples were clustered and loaded equally over two lanes on an SP flow cell on an Illumina Nova-Seq 6000 Sequencing system for 50bp paired end reads.

## RNA sequencing analysis

Raw sequencing reads were pseudoaligned to human reference transcriptome GRCh38.p13 using Kallisto [28] version 0.45.0, and a gene-by-sample count matrix was generated for differential gene expression analysis with iDEP v.0.92 [29]. Next, up- and downregulated genes (p≤0.10) from our calcitriol supplemented hAECs were compared with human AMP-related genes putatively involved in a variety of innate immune response, and human genes involved in motile cilium using QuickGO annotations [30].

## Neutrophil migration across hAECs

Gene expression data revealed *TREM1* to be highly upregulated in hAECs. *TREM1* is abundantly expressed on human myeloid cells, and facilitates neutrophil migration across hAECs [18], therefore, we evaluated the role of calcitriol on neutrophil migration across hAECs *in vitro*.

Bronchial hAECs were obtained from human donor lungs, n = 4 (donors were on average 40 years old (SD = 18), and 50% female) and seeded at the ALI onto 3 μm-pore-size collagen-coated transwell filters (Corning Life Sciences, Durham, NC, USA) in an inverted fashion with basolateral support provided by growth-factor reduced Matrigel® Matrix basement

membrane (Corning Life Sciences, Durham, NC, USA). Upon hAEC polarization (2 weeks), calcitriol (100 nM) was added to the basolateral media in half of the wells.

The following day (24h), fresh blood was obtained from one healthy volunteer donor, and neutrophils were isolated and suspended in HBSS-/- (Gibco), as previously described (with modifications) [31]. Next, hAEC Matrigel was removed and transwells were inverted into 400 μL USG, then neutrophils were applied to the basolateral membrane (200,000/well) to assess their migration to the apical membrane. Half of the transwells were suspended in 30% zymosan activated serum, ZAS, a chemoattractant for neutrophils [18]. After three hours, migrated neutrophils were collected from the plate using PBS -/-, 2mM EDTA, and TrypLE (Gibco/Invitrogen, Waltham, MA, USA). Migrated cells were stained with fixable aqua dead cell stain kit (Thermo Fisher Scientific, Waltham, MA, USA), Human BD Fc Block™ (BD Biosciences, San Jose, CA, USA), and APC anti-human CD66b antibody (Biolegend, San Diego, CA, USA). CD66b+ cells assessed by flow cytometry using an Attune™ NxT Flow Cytometer Autosampler (Invitrogen™/Thermo Fisher Scientific, Waltham, MA, USA). CD66b+ cells were subsequently analyzed using FloJo™ software (BD Biosciences, San Jose, CA, USA). Each condition was performed in triplicate and significance determined by one-sample *t*-test (theoretical mean = 100) of baseline-transformed replicate data by experiment.

## Results

### ASL antimicrobial and antiviral activity

Compared to control, untreated ASL killed *S. aureus* and ASL from hAECs treated with 100 nM calcitriol (for 24h) increased *S. aureus* killing (mean live *S. aureus*, untreated ASL = 40%, mean live *S. aureus*, treated ASL = 27%, one-sample t-test with hypothetical mean = 100, $p<0.001$ both conditions). ASL from supplemented hAECs was significantly more bactericidal than ASL from unsupplemented hAECs (paired t-test, p = 0.02), **Fig 1A**.

ASL from both conditions significantly reduced 229E-CoV infectivity compared to vehicle (viral plaques with untreated ASL $p<0.01$ and with supplementation $p<0.05$). Calcitriol supplementation did not affect ASL-mediated 229E-CoV inactivation (paired t-test, p = 0.39), **Fig 1B**.

### Effect of calcitriol supplementation on hAEC ASL pH, viscosity, and ciliary beat frequency

ASL pH from hAECs treated with basolateral calcitriol supplementation was not significantly different than the ASL pH from untreated control hAECs ($p = 0.11$), **Fig 2A**. Acute and 24-hour ASL viscosity did not change after calcitriol treatment (0 vs. 45 min $p = 0.98$, 0 vs. 90 min $p = 0.20$, 0 vs. 24h, $p = 0.36$), **Fig 2B**. Calcitriol supplementation did not affect CBF of hAECs acutely nor over 24h (0 vs. 45 min $p = 0.98$, 0 vs. 90 min $p = 0.20$, 0 vs. 24h, $p = 0.36$), **Fig 2C**.

### Effect of calcitriol supplementation on hAEC gene expression

We assessed gene expression changes to hAECs after 24h calcitriol supplementation. The top ten significantly upregulated genes in calcitriol treated hAECs (compared to controls) were the following: *CYP24A1*, *CAMP*, *PLG*, *HRCT1*, *SULT1C2*, *CA9*, *PRKG2*, *TREM1*, *IL1RL1*, and *PADI3* (all $p<0.001$). Compared to control epithelia, the following genes were the most downregulated in calcitriol treated hAECs: *IL19*, *IGF2*, *PRRT1B*, *MUC13*, *SLC5A5*, *LTK*, *SLURP2*, *DEPP1*, *LAMB1*, and *SLC9A4* (all $p<0.05$).

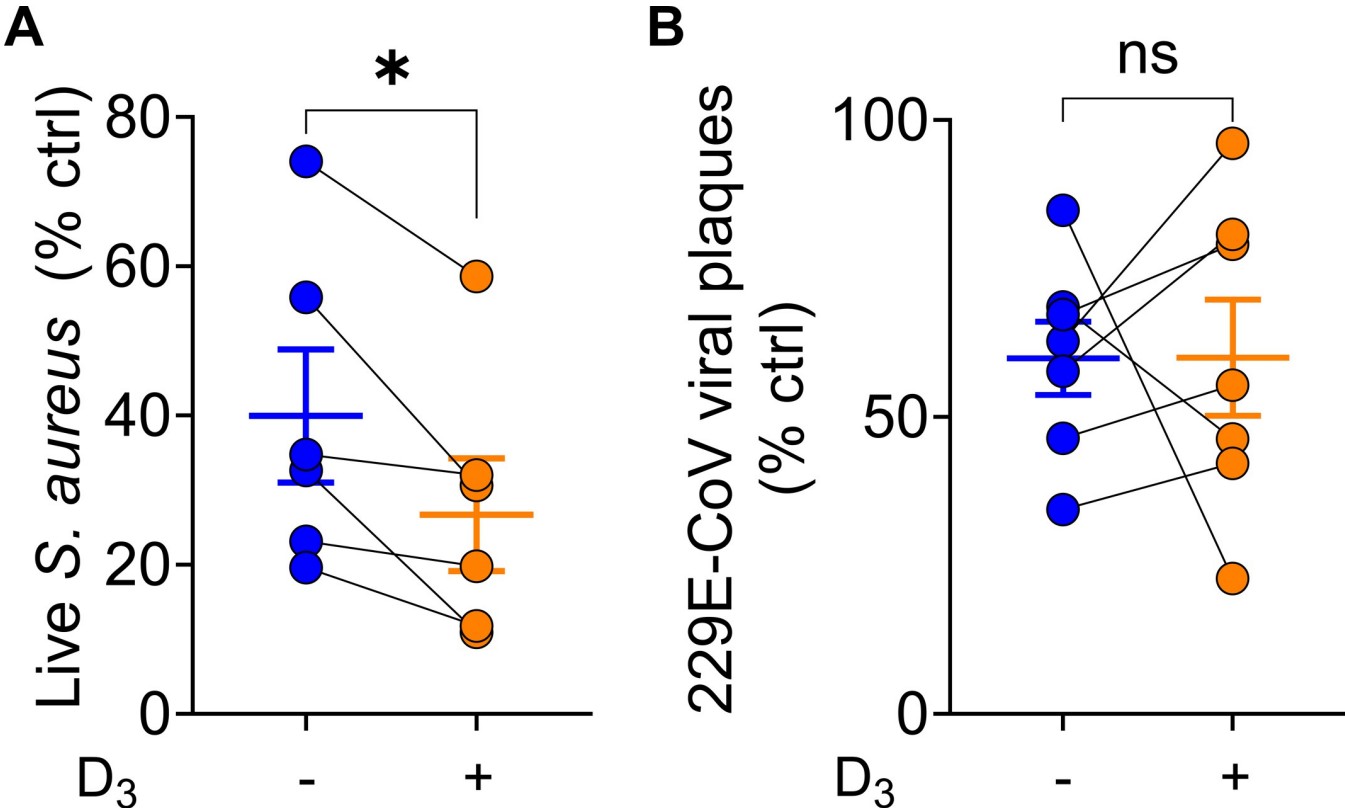

**Fig 1. ASL bacterial and viral inactivation from paired hAECs untreated and treated with calcitriol.** A. ASL from untreated (blue dots) and calcitriol-treated (orange dots) hAECs or vehicle control was challenged with bioluminescent *S. aureus* and RLUs were read within 12 minutes; individual points represent mean data from six individual experiments, ASL derived from 4–6 individual hAEC donors, * indicates $p<0.05$ B. ASL from untreated (blue dots) and calcitriol-treated (orange dots) hAECs compared to vehicle control. Conditions were challenged with 229E-CoV in MRC-5 human fibroblasts, and viral plaques quantified. Individual points represent mean data from seven individual experiments, ASL was derived from 4–6 individual hAEC donors; raw plaque values for control, untreated hAEC ASL, and supplemented hAEC ASL were compared by a paired Friedman test, ASL vs. vehicle control $p<0.01$, ASL from hAECs supplemented with calcitriol vs. vehicle control $p = 0.049$, treated vs. untreated ASL $p>0.99$. All studies were performed in triplicate and error bars represent mean± SEM.

Because AMP expression and composition affect ASL antiviral and antibacterial activity, we compared our RNA sequencing results with AMP-related genes involved in innate immune response ($p\leq0.10$) from a previously published database [30]. Results from genes involved in a wide-range of host-defense are reported, including those involved in overall innate immune response, defense response to virus, type 1 interferon signaling, defense response to gram negative and positive bacterium, and antifungal response, as well as others. AMP-related genes most upregulated in our sample are shown in **Table 1A**, and those most downregulated can be seen in **Table 1B**.

Using the same protocol, we compared genes involved in human motile cilium against hAEC genes affected by supplementation. *FSIP2*, *HIF1A* and *CABYR* were upregulated (log2 fold-change = 0.53; $p = 0.075$, 0.36; $p = 0.006$, and 0.26; $p = 0.064$, respectively) and *ATP1B1* was downregulated (log2 fold-change = 0.39; $p = 0.009$). These genes' roles include motile cilium and sperm components.

Because calcitriol alters the expression of genes that regulate $Ca^{2+}$, we compared genes associated with intracellular $Ca^{2+}$ in our sample. *TMEM37*, *TRPV6*, and *CHRNA7* were significantly upregulated (log2 fold-change = 1.43; $p<0.001$, 0.802; $p<0.001$, and 0.776; $p<0.001$, respectively), while *CACNB1* was significantly downregulated (log2 fold-change = 0.567; $p<0.001$).

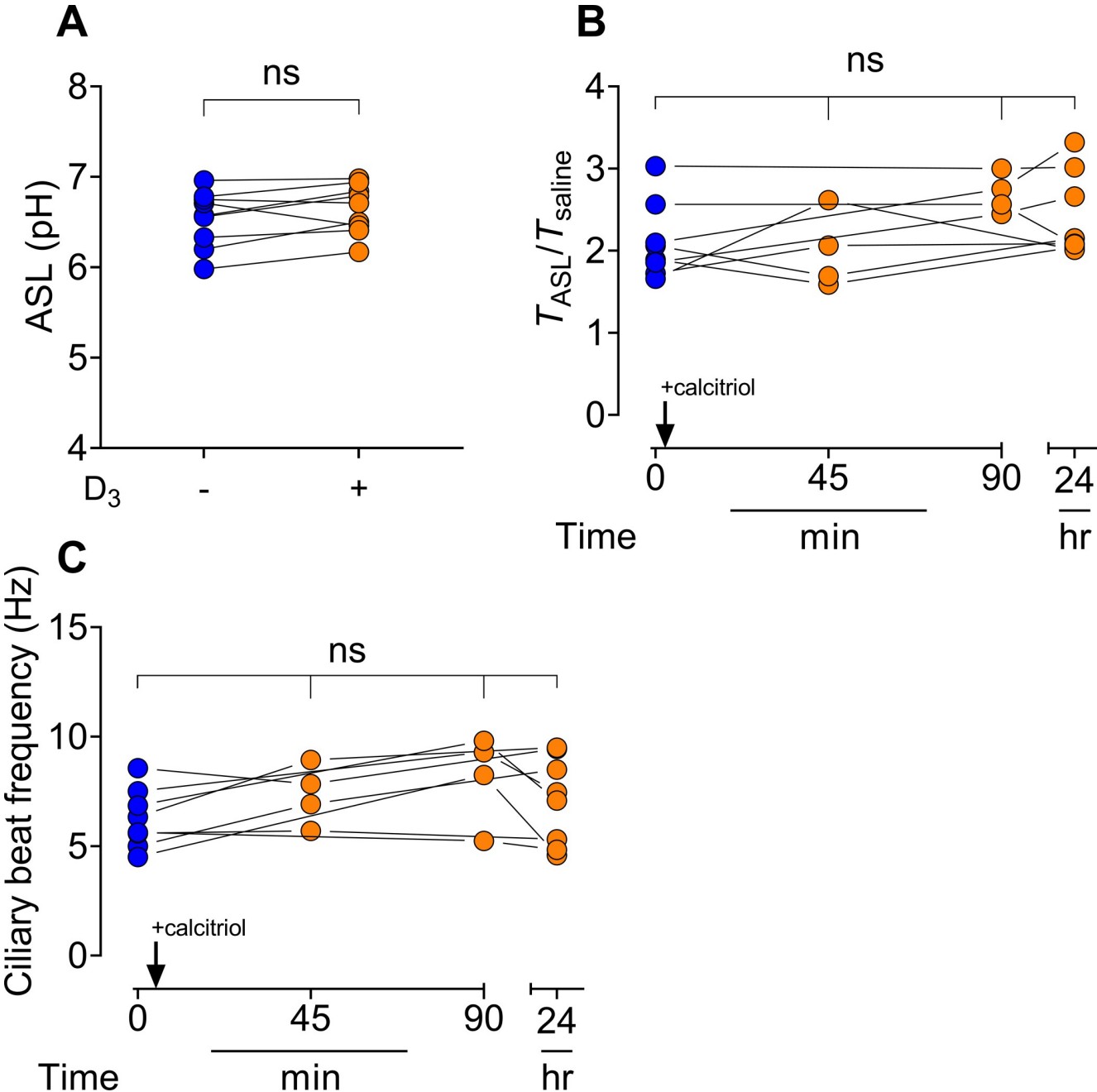

**Fig 2. ASL pH and viscosity from hAECs treated without and with calcitriol. A.** Calcitriol free and supplemented basolateral hAEC media (n = 9 donors) was changed when apical saline+1mM HEPES solution was applied; after 22h ASL was collected, and pH read. Significance determined by paired *t*-test, p = 0.11. **B.** FITC-D within ASL (≥12h) was excited (488 nm), baseline images were acquired then a small circular region (radius 5.2 μm) was photobleached and fluorescent recovery monitored for >1 min; recovery was compared to FITC-D in PBS. Significance determined by one-way ANOVA with Šídák's multiple comparisons test, 0 vs. 45 min $p = 0.98$, 0 vs. 90 min $p = 0.20$, 0 vs. 24h, $p = 0.36$. **C.** Time-course of paired (n = 4 or n = 8 per timepoint) hAEC CBF acquired using brightfield imaging before (0 min) and after 100 nM basolateral calcitriol supplementation, significance determined by one-way ANOVA with Šídák's multiple comparisons test, 0 vs. 45 min $p = 0.40$, 0 vs. 90 min $p = 0.20$, 0 vs. 24h, $p = 0.39$.

### Neutrophil migration across hAECs

Because *TREM1* was significantly upregulated in our calcitriol-treated sample, we assessed the effect of calcitriol supplementation on neutrophil migration across hAECs at the ALI.

**Table 1. Effect of calcitriol treatment on hAEC antimicrobial protein gene expression.**

| A | | | B | | |
|---|---|---|---|---|---|
| Gene | FC (log2) | *p* | Gene | FC (log2) | *p* |
| *CAMP* | 7.36 | <0.001 | *UBD* | 0.97 | <0.001 |
| *TREM1* | 2.90 | <0.001 | *CCL17* | 0.80 | 0.009 |
| *CD14* | 1.61 | <0.001 | *RNASE7* | 0.71 | 0.089 |
| *SELP* | 1.52 | <0.001 | *CCL20* | 0.68 | 0.075 |
| *SLFN11* | 1.22 | <0.001 | *VNN1* | 0.67 | 0.008 |
| *KRT16* | 1.17 | <0.001 | *APOBEC3B* | 0.65 | 0.062 |
| *LGALS9* | 0.80 | <0.001 | *ZAP70* | 0.61 | 0.012 |
| *RNF39/PPP1R11* | 0.64 | <0.001 | *TGFBI* | 0.58 | 0.098 |
| *SLC15A2* | 0.54 | <0.001 | *CIITA* | 0.53 | <0.001 |
| *RARRES2* | 0.51 | 0.037 | *PGLYRP4* | 0.52 | 0.014 |
| *RIPK2* | 0.51 | 0.001 | *NLRP1* | 0.49 | 0.073 |
| *CH25H* | 0.49 | <0.001 | *IL22RA1* | 0.47 | 0.074 |
| *ZYX* | 0.46 | 0.007 | *CX3CL1* | 0.43 | 0.017 |
| *CLU* | 0.45 | <0.001 | *SCNN1B* | 0.41 | 0.076 |
| *SLAMF7* | 0.43 | 0.033 | *IFITM2* | 0.37 | 0.020 |
| *KYNU* | 0.43 | <0.001 | *SPON2* | 0.36 | 0.016 |
| *RPL30* | 0.34 | 0.096 | *GBP2* | 0.36 | 0.064 |
| *H2BC4* | 0.29 | 0.062 | *DAPK1* | 0.35 | 0.004 |
| *TRIM38* | 0.28 | <0.001 | *RAET1E* | 0.32 | 0.002 |
| *CEBPB* | 0.27 | 0.025 | *OAS3* | 0.31 | 0.094 |
| *RAB20* | 0.25 | 0.073 | *FLNB* | 0.30 | 0.018 |
| *SP100* | 0.24 | 0.001 | *CLDN1* | 0.30 | 0.006 |
| *TBK1* | 0.24 | 0.018 | *TRIM29* | 0.30 | 0.001 |
| *ELF4* | 0.23 | 0.031 | *OAS1* | 0.26 | 0.052 |
| *TRIM35* | 0.23 | 0.064 | *STING1* | 0.24 | 0.006 |
| *TRAF4* | 0.19 | 0.095 | *TRIM14* | 0.24 | 0.021 |
| *HK1* | 0.18 | 0.098 | *CD74* | 0.23 | 0.043 |
| *F2RL1* | 0.17 | 0.088 | *OTULIN* | 0.18 | 0.070 |
| | | | *ZNFX1* | 0.16 | 0.046 |

**A.** Gene expression of AMP-related genes upregulated in calcitriol-treated hAECs compared to paired control hAECs, expressed as log2 fold change, displayed in descending order of magnitude **B.** Gene expression of AMP-related genes downregulated in calcitriol-treated hAECs compared to paired control hAECs, expressed as log2 fold change, displayed as absolute values, in descending order of magnitude.

Overnight calcitriol treatment did not significantly affect neutrophil migration ($p = 0.39$), while ZAS increased neutrophil migration regardless of basolateral hAEC supplementation ($p<0.05$). Under ZAS supplementation, neutrophil migration tended to be higher in calcitriol-treated cells (paired *t*-test, $p = 0.09$) compared to untreated cells.

## Discussion

Vitamin D supplementation may reduce the risk of respiratory infection and improve airway innate immunity, but mechanisms supporting its effect are not well understood. We systematically tested the effect of calcitriol supplementation on key innate immune mechanisms. We found that calcitriol supplementation of hAECs increased ASL antibacterial activity but did not increase its antiviral activity. Within our calcitriol-treated sample, cathelicidin (*CAMP*) was the second most upregulated gene, which likely explains the observed increase in

antibacterial activity. Calcitriol treatment of hAECs did not impact ASL pH or viscosity but did acutely affect CBF. While *TREM1* was overexpressed in our calcitriol-treated sample, we did not find an appreciable effect on neutrophil migration across hAECs *in vitro*.

Consistent with previous literature, calcitriol supplementation of hAECs led to significantly increased cathelicidin (*CAMP*/LL-37) expression (log2 FC = 7.36, *p*<0.001), **Table 1A** [32–34]; cathelicidin is one of many AMPs within the ASL [12, 20, 35, 36], and its expression has been shown to increase antibacterial (*Mycobacterium tuberculosis*) activity after calcitriol supplementation [14]. Additionally, silencing *CAMP* gene expression, and blocking LL-37, inhibits the cathelicidin-induced increase in antimicrobial activity [14, 37]. Therefore, the increased ASL-mediated *S. aureus* killing in calcitriol treated hAECs observed in our study is likely due to increased cathelicidin expression. However, given the abundance of AMP-related gene expression alterations, we cannot solely ascribe this finding to cathelicidin.

Antimicrobial bacterial killing is affected by ASL pH [38]. We did not find a difference in ASL pH between calcitriol-treated and untreated hAECs. Additionally, increased ASL viscosity has been observed in certain lung disease (e.g. cystic fibrosis), and viscosity changes can occur independently of ASL pH [39]; however, we found hAEC ASL viscosity after treatment with calcitriol was not significantly different compared to pre-treatment viscosity (*p*>0.20), **Fig 2B**. Next, we tested whether ciliary movement was altered upon calcitriol supplementation. Epithelial cilia work in a coordinated fashion to move incoming pathogens toward the larynx, and increased CBF abets this process. Changes to CBF may be due to activation of cellular signaling pathways (acutely) and/or alterations to gene expression (hours-days). As seen in **Fig 2C**, activation of the vitamin D receptor did not significantly change CBF over time. While CBF varied stochastically, it typically remained higher than prior to supplementation for most donors. This variation in CBF may result from gene expression-dependent (*e.g.*, upregulation of *FSIP2*, *HIF1A* and *CABYR* or downregulation of *ATP1B1*) or gene expression-independent changes in cilia regulation. Fluctuations in CBF were observed within each set of experiments.

It has been proposed that vitamin D deficiency increases the risk of COVID-19 incidence and severity [40]. We recapitulated our previous work demonstrating ASL to be virucidal [41], however, we found no difference in ASL-mediated 229E-CoV viral inactivation between calcitriol-treated and untreated hAEC ASL. This is consistent with a recent finding from the United Kingdom where blood 25(OH)D concentrations did not correlate with COVID-19 infection risk [42]. However, vitamin D supplementation as a means to reduce respiratory infection in clinical trials yields inconsistent results [43]. In one retrospective cohort study (n = 489), likely vitamin D deficiency was imputed based on recent 25(OH)D$_3$ or calcitriol serum levels and treatment and likely deficient vitamin D levels were associated with an increased rate of COVID-19 positivity [44]. *In vitro* work revealed that calcitriol treatment of hAECs can reduce inflammatory signaling after exposure to viral influenza (H1N1) infection, reducing autophagy and returning apoptosis to constitutive levels [22]. While evidence supports a role for vitamin D supplementation and reduced risk of viral infection, we focused on strictly innate immune ASL-mediated viral inactivation and found this not to be influenced by supplementation.

RNA sequencing data revealed calcitriol supplementation led to significantly increased *TREM1* expression. Previous work has demonstrated increased *TREM1* expression in calcitriol-treated hAECs from distinct cell lines [45], however, to our knowledge, ours is the first study to identify *TREM1* expression in endogenous primary hAECs. *TREM1* is a membrane-bound surface receptor with an unknown ligand, while its soluble form may act as a counterregulatory decoy receptor [46]. Because *TREM1* is required for neutrophil migration across hAECs, we expected increased neutrophil migration in calcitriol-treated hAECs. We did not observe an increase in neutrophil migration after hAECs were treated with calcitriol, however, **Fig 3**. Zymosan activated serum, a neutrophil chemokine was used as a positive control and

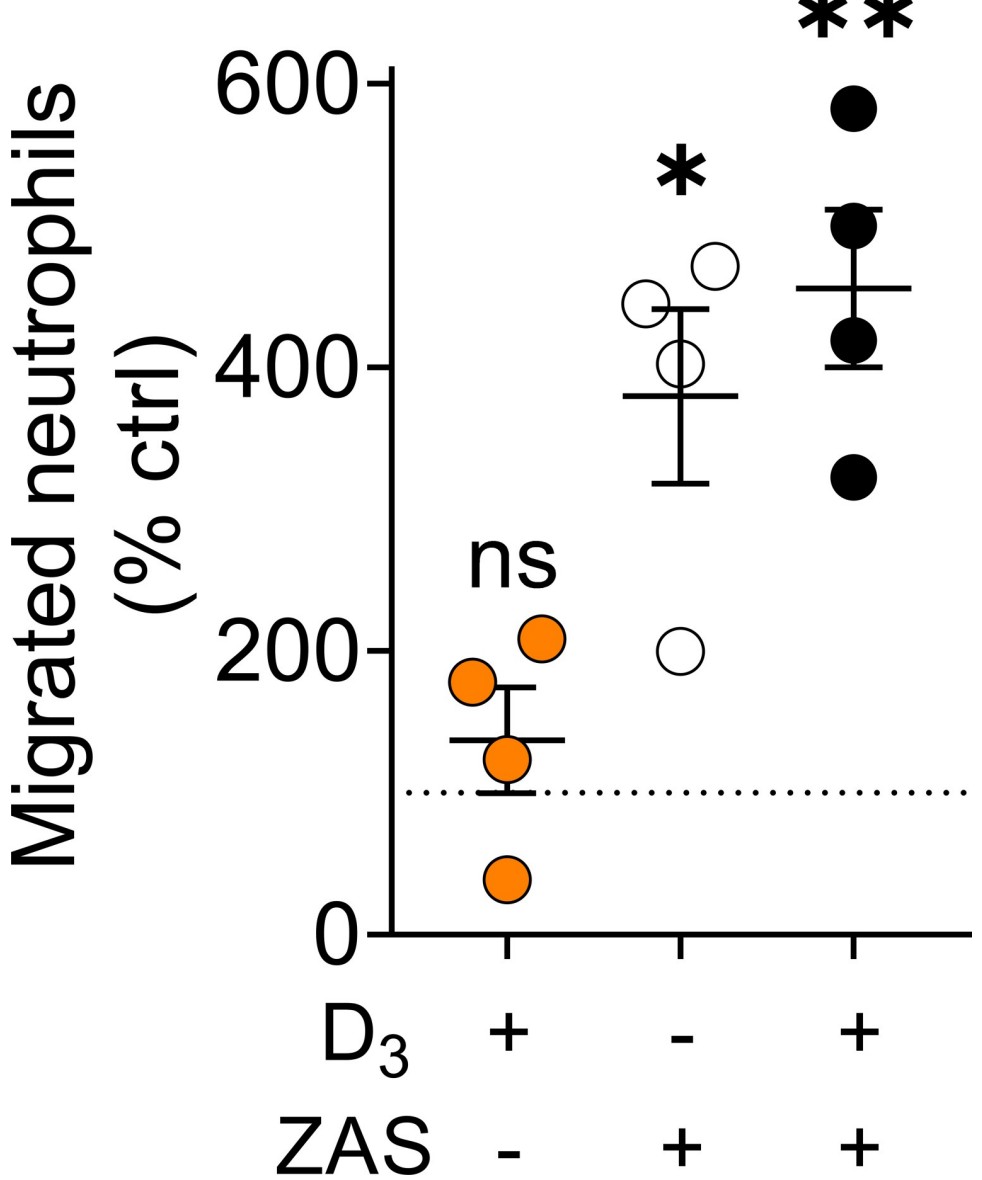

**Fig 3. Effect of overnight calcitriol treatment on neutrophil migration across hAECs.** Percent of migrated neutrophils across hAECs with (orange and black circles) and without (white circles) overnight calcitriol treatment, n = 4 hAEC donors, each dot represents the average of triplicate data per experiment, neutrophils isolated from one healthy donor. Neutrophil migration was performed under USG and 30% zymosan activated serum (ZAS) conditions; significance determined by comparing baseline-corrected data of each condition to untreated hAECs using a one-sample t-test with hypothetical mean of 100, * indicates p<0.05, ** indicates p<0.01, error bars indicate mean ±SEM.

significantly increased neutrophil migration. We observed a slightly increased migration upon supplementation with ZAS (paired *t*-test *p* = 0.09) but did not find significant differences between control and calcitriol treated cells. This may indicate calcitriol supplementation can enhance migration in the presence of yeast pathogens.

Our study has several limitations. All studies were performed *in vitro* and extrapolation to *in vivo* effects is therefore limited. Our *in vitro* design relied on airway epithelia from organ donors whose vitamin D status was unknown at the time of organ harvest. Therefore, we are

unable to speculate whether the epithelial innate immune response to vitamin D supplementation observed in our study was from donors with healthy or deficient vitamin D levels. Further, our calcitriol supplementation dose of 100 nM may be significantly greater than that found in healthy serum (~200 picomolar) [47], but we are unable to measure endogenous calcitriol in human airway epithelia and therefore selected the dose previously assayed with human airway cells [20–22]. For any given experiment, at least four donors were studied, so it is unlikely, but still possible, that all were deficient. To reduce variability in response to the vitamin D receptor, our experiments utilized active vitamin D $(1,25(OH)_2D_3)$, thus our results cannot be extrapolated to the stimulation of cells with inactive vitamin D $(25(OH)_2D_3)$. Our experimental design relied on supplementation $\leq$24 hours, and therefore effects from longer stimulation are unknown. We did not evaluate synergistic and additive effects of AMPs. Our antimicrobial results are limited to ASL effects on one type of relevant bacteria and virus, and we cannot extrapolate these towards all pathogen species, including other bacteria, viruses, fungal and mycobacterial agents. For neutrophil migration assays, only hAECs were stimulated and not neutrophils, which may have led to effects on migration. Finally, we used primary hAECs and our results cannot be extrapolated to other cell types, including alveolar epithelial cells.

Despite these limitations, our findings demonstrate that calcitriol treatment increases ASL bactericidal properties (*S. aureus*) and significantly alters hAEC gene expression, especially AMP-related genes, while calcitriol does not affect ASL-antiviral (229E-CoV) properties, pH, viscosity or CBF. In addition, neutrophil migration may be increased if chemokines are present in the air space. Vitamin D supplementation to increase active vitamin D concentrations may provide a useful tool to resolve respiratory bacterial infections.

## Supporting information

**S1 Fig. Relative light units of control conditions from S. aureus bacterial killing assays.** Control (vehicle) media and *S. aureus* resulted in a slight decline of bacteria over 12 minutes, while bleach and *S. aureus* led to complete bacterial killing over the same time.
(DOCX)

**S2 Fig. *S. aureus* growth in media with and without calcitriol supplementation.** A clinically isolated strain of methicillin-resistant *S. aureus* was cultured overnight in tryptic soy broth (TSB), then sub-cultured (1:100) in TSB +/- $10^{-7}$ M calcitriol. Paired t-tests of % growth compared to time 0 (at hours 1–6) demonstrated no statistically significant differences between control and supplemented media (all *p*>0.05). This experiment was independent of airway mechanisms.
(DOCX)

## Acknowledgments

We would like to thank Sankar Baruah at the University of Iowa for his *TREM1* assistance and trouble shooting. We would also like to thank Christian Zirbes for providing us with an *S. aureus* clinical isolate for our bacterial growth experiments, and for his prompt and helpful assistance. The University of Iowa Cell Culture Core provided invaluable assistance in isolating primary human donor cells and providing hAECs at the ALI.

## Author Contributions

**Conceptualization:** Julia Klesney-Tait, Alejandro P. Comellas.

**Data curation:** Emma M. Stapleton, Andrew L. Thurman, Alejandro A. Pezzulo.

**Formal analysis:** Emma M. Stapleton, Andrew L. Thurman, Ian M. Thornell, Alejandro A. Pezzulo.

**Funding acquisition:** Joseph Zabner.

**Investigation:** Emma M. Stapleton, Kathy Keck, Robert Windisch, Mallory R. Stroik, Ian M. Thornell.

**Methodology:** Mallory R. Stroik, Ian M. Thornell, Alejandro A. Pezzulo, Julia Klesney-Tait, Alejandro P. Comellas.

**Project administration:** Alejandro P. Comellas.

**Resources:** Joseph Zabner, Julia Klesney-Tait.

**Validation:** Alejandro P. Comellas.

**Writing – original draft:** Emma M. Stapleton.

**Writing – review & editing:** Emma M. Stapleton, Kathy Keck, Andrew L. Thurman, Joseph Zabner, Ian M. Thornell, Alejandro A. Pezzulo, Julia Klesney-Tait, Alejandro P. Comellas.

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
