## [Decision Letter · Decision Letter 0]

12 Apr 2022

PONE-D-22-05766Vitamin D-mediated effects on airway innate immunity in vitroPLOS ONE

Dear Dr. Comellas,

Thank you for submitting your manuscript to PLOS ONE. After careful consideration, we feel that it has merit but does not fully meet PLOS ONE’s publication criteria as it currently stands. Therefore, we invite you to submit a revised version of the manuscript that addresses the points raised during the review process.

Please address the reviewers' comments. In addition, please clarify if RNAseq data will be deposited and accessible to other researchers. Many of the raised concerns with methodology, interpretation, and statistic analysis could be addressed by clarification. However, as suggested by reviewer 2, results from a control experiment regarding the effect of experimental concentration of Vitamin D alone on S. aureus strain should be included.

We look forward to receiving your revised manuscript.

Kind regards,

Y. Peter Di, Ph.D.

Academic Editor

PLOS ONE

Journal Requirements:

Reviewers' comments:

Reviewer's Responses to Questions

**Comments to the Author**

1. Is the manuscript technically sound, and do the data support the conclusions?

Reviewer #1: Yes

Reviewer #2: Partly

2. Has the statistical analysis been performed appropriately and rigorously? 

Reviewer #1: Yes

Reviewer #2: I Don't Know

3. Have the authors made all data underlying the findings in their manuscript fully available?

Reviewer #1: Yes

Reviewer #2: No

4. Is the manuscript presented in an intelligible fashion and written in standard English?

Reviewer #1: Yes

Reviewer #2: Yes

5. Review Comments to the Author

Reviewer #1: The experiments examining the effects of calcitriol (vitamin D) on the antibacterial and antiviral effects of the airway surface liquid from primary human airway cultures are straightforward and clear. They indicate that calcitriol does enhance antibacterial (S. aureus) but not antiviral (22lE-CoV) effects in this primary cell culture system. Calcitriol also had no effect on the airway surface liquid pH or viscosity, or the ciliary beat frequency. The expression of a few genes was altered and the results of those alterations may be interesting

There is a single, very small Supplementary table describing the sources of the airway culture donors that should be included in the body of the manuscript in the Materials and methods.

Reviewer #2: This study examines the effect of calcitriol supplementation on airway epithelial cells in vitro. The literature surrounding the role of calcitriol/vitamin D in respiratory infections/immunity is somewhat murky, and this study is a worthwhile contribution to the field. I complement the authors on an interesting and largely well written paper, but I have some questions regarding methods, controls, and interpretation that I believe need to be addressed prior to the paper being acceptable for publication.

(1) The stats are very confusing in this paper. In the first paragraph of the results (line 160) describing one A the authors say “one-sample t-test with hypothetical mean=100, p<0.001 both conditions)” but the figure has one asterisk between the two groups. The methods talks about paired t tests (lines 87, 93, 99, etc.). Then line 164 reports ANOVA for 1B (which is only a 2 group comparison). Use of stats seem to be a little all over the place and somewhat inconsistent.

(2) There’s no question that the data the authors show here supports an effect of 100 nM calcitriol on parameters of innate immunity. My main concern is the interpretation of the results. There is (as the authors imply in the first paragraph of the introduction) a difference between the potential usefulness of vitamin D as a supplement vs the role of vitamin D deficiency in respiratory infection susceptibility. I guess my main question with this study is which condition does this study truly model? Are the ALIs at baseline *deficient* in vitamin D or does their baseline reflect a normal healthy level and the authors are just getting the ALIs to normal? I know this is a subtle distinction but it’s important for thinking about what the results here might mean in the context of in vivo infection. Some discussion of this in the discussion section would be useful.

(3) The first sentence of the methods says “We selected our calcitriol dose based on prior work…” Since there are no limits in this journal could the authors please say a few sentences to briefly discuss this? It would help with interpretation of the study without forcing the reader to find another reference. That’s a key component of the study. Is 100 nM calcitriol something achievable with supplementation? I understand this may be addressed in the referenced paper but because this is so important for interpretation, there needs to be at least some discussion of the authors’ rationale in this paper.

(4) The description of the media might be incomplete (methods ln 74) as many labs’ homemade ALI medias contains more than just 2% Ultroser G. Is there any baseline calcitriol or other vitamin D form in the Ultrosser G?

(5) Fig 1: There’s no control as to whether D3 alone has effects on bacterial growth along. Growth should be tested in media ± D3 alone. This is a pretty easy control that would help the interpretation of the results from panel A.

(6) Fig 2: Panel A has a n.s. p value reported while the other panels have “ns”. Pick one style. It doesn’t make the results in panel A better than the others because there’s a P value somewhat closer to p<0.05. Either report all the NS p values or none.

(7) Line 113 refers to supplemental methods but I don’t see any supplemental methods

(8) More description of the ciliary beat frequency measurements is needed to determine if the authors are appropriately powered to measure CBF. What is the frame rate? Are they at or greater than Nyquist sampling rate? Is this single line scanning (they say it was done on a Zeiss LSM880)? More description of this is needed here. Just want to make sure whenever someone reports no change in CBF that they are imaging fast enough to see the changes if there are changes.

6. PLOS authors have the option to publish the peer review history of their article (what does this mean?). If published, this will include your full peer review and any attached files.

Reviewer #1: No

Reviewer #2: No

---

## [Author Response · Author response to Decision Letter 0]

22 Apr 2022

Reviewer #1: The experiments examining the effects of calcitriol (vitamin D) on the antibacterial and antiviral effects of the airway surface liquid from primary human airway cultures are straightforward and clear. They indicate that calcitriol does enhance antibacterial (S. aureus) but not antiviral (22lE-CoV) effects in this primary cell culture system. Calcitriol also had no effect on the airway surface liquid pH or viscosity, or the ciliary beat frequency. The expression of a few genes was altered and the results of those alterations may be interesting

Comment: There is a single, very small Supplementary table describing the sources of the airway culture donors that should be included in the body of the manuscript in the Materials and methods.

Response: This is a great observation. We completely agree and have updated our Neutrophil migration across hAECs methodology to include the text: “donors were on average 40 years old (SD=18), and 50% female.”

Reviewer #2: This study examines the effect of calcitriol supplementation on airway epithelial cells in vitro. The literature surrounding the role of calcitriol/vitamin D in respiratory infections/immunity is somewhat murky, and this study is a worthwhile contribution to the field. I complement the authors on an interesting and largely well written paper, but I have some questions regarding methods, controls, and interpretation that I believe need to be addressed prior to the paper being acceptable for publication.

Comment (1): The stats are very confusing in this paper. In the first paragraph of the results (line 160) describing one A the authors say “one-sample t-test with hypothetical mean=100, p<0.001 both conditions)” but the figure has one asterisk between the two groups. The methods talks about paired t tests (lines 87, 93, 99, etc.). Then line 164 reports ANOVA for 1B (which is only a 2 group comparison). Use of stats seem to be a little all over the place and somewhat inconsistent.

Response (1): We thank the reviewer for their close reading and for bringing this to our attention. We clarified which tests were used for which reasons in different assays. We streamlined the results section to be sure not to over-report unnecessary statistical analyses. 

For the first example (“one-sample t-test with hypothetical mean p<0.001”), the text and the figure legends were referring to two distinct statistical analyses. The one-sample t-test with hypothetical mean=100 was performed to assess whether bacterial killing occurred with either condition (ASL+/- D), but the figure legend indicated a paired t-test to test between conditions (+/- D). To clarify, we updated the first paragraph of the results text to reflect that our paired t-test (and single asterisk between groups) was to test between conditions (ASL killing +/-D), with a paired reference to the figure legend. 

We have updated the methods to more accurately report which statistical tests were run in each assay.

We thank the reviewer for noting our discrepancy in line 164 reporting an ANOVA for 1B. This was a complete error – we had performed a paired t-test (as noted in the methods section for ASL antiviral activity against 229E-CoV). We have updated the text accordingly and thank the reviewer.

Our use of one-sample t-test with hypothetical mean=100 in Fig 3 was due to biological variability within our control condition (neutrophils behave differently on different days and migrate differently in distinct human donor epithelia).

(2) There’s no question that the data the authors show here supports an effect of 100 nM calcitriol on parameters of innate immunity. My main concern is the interpretation of the results. There is (as the authors imply in the first paragraph of the introduction) a difference between the potential usefulness of vitamin D as a supplement vs the role of vitamin D deficiency in respiratory infection susceptibility. I guess my main question with this study is which condition does this study truly model? Are the ALIs at baseline *deficient* in vitamin D or does their baseline reflect a normal healthy level and the authors are just getting the ALIs to normal? I know this is a subtle distinction but it’s important for thinking about what the results here might mean in the context of in vivo infection. Some discussion of this in the discussion section would be useful.

Response (2): We very much agree with the reviewer that the vitamin D status of donors within our model may provide an important insight into the observed responses. This is an interesting distinction. Unfortunately, we rely on organ donors whose vitamin D status is unknown to us upon harvest. We have tried to clarify this point in the limitations section of the Discussion. We will consider this for future work.

(3) The first sentence of the methods says “We selected our calcitriol dose based on prior work…” Since there are no limits in this journal could the authors please say a few sentences to briefly discuss this? It would help with interpretation of the study without forcing the reader to find another reference. That’s a key component of the study. Is 100 nM calcitriol something achievable with supplementation? I understand this may be addressed in the referenced paper but because this is so important for interpretation, there needs to be at least some discussion of the authors’ rationale in this paper.

Response (3): This is an important observation. We based our calcitriol dose on that which was consistently selected in previous work evaluating effects of calcitriol supplementation on human airway cells (see Hansdottir et al., 2010; Banerjee et al., 2009; Khare et al., 2012; Gui et al. 2017). Measurements of calcitriol in human serum are evolving. Semi-recent work shows healthy human serum to be 100-200 picomolar (calculated from Souberbielle et al. 2015) – as the reviewer can appreciate, our dose is significantly greater than this (103). We have added this to the limitation paragraph in our discussion section and have added additional citations and text to our methodology section. Unfortunately, it is very difficult to know the true in vivo concentration of basolateral calcitriol.

(4) The description of the media might be incomplete (methods ln 74) as many labs’ homemade ALI medias contains more than just 2% Ultroser G. Is there any baseline calcitriol or other vitamin D form in the Ultrosser G? 

Response (4): We thank the reviewer for this inquiry. Our lab does not add calcitriol to the media. Additionally, we communicated with the USG manufacturer and were told that while the recipe is proprietary, it contains recombinant Fibroblast Growth Factor-Basic, progesterone, estradiol, testosterone, tri-iodothyronine and dexamethasone. Therefore, to our knowledge, our hAECs at the ALI are not supplemented with any form of vitamin D. Because this was a private communication with the retailer, we do not want to share their recipe in the manuscript, but we have added the following sentence to the methods section:

“The exact composition of USG is confidential, but the manufacturer correspondence indicates it does not contain calcitriol.”

(5) Fig 1: There’s no control as to whether D3 alone has effects on bacterial growth along. Growth should be tested in media ± D3 alone. This is a pretty easy control that would help the interpretation of the results from panel A.

Response (5): This is a good observation. We tested the effect of immediate (within 10 minutes) bacterial killing in minimal media (alongside S. aureus (negative) control and S. aureus + bleach (positive) control) and compared killing to each condition’s time zero. Below is a graph of control conditions from the bacterial experiments included in our manuscript. We do not feel these controls contribute meaningful content to our manuscript, but we are now including them in supplementary data – as can be appreciated, S. aureus did not grow in the short timeframe of our assay (it slightly decreased likely due to use of minimal media, but ASL +/- D decreased more). Bleach effectively killed all S. aureus within 10 minutes. 

Due to this reviewer comment (“growth should be tested in media +/- D3 alone”), we tested whether vitamin D independently affects bacterial growth. Because bacteria don’t grow in the timespan required in the bacterial killing assay, we decided to test whether 100 nM calcitriol supplementation affected S. aureus growth over 5 hours in rich media (100% TSB), +/- 10-7 calcitriol, then assessed optical density (OD). Conditions were performed in parallel, and each condition was run in triplicate over three separate days. Because we have no way of knowing how much of the basolaterally supplemented calcitriol accumulates in the apical ASL, we are adding this growth experiment to the supplementary data.

(6) Fig 2: Panel A has a n.s. p value reported while the other panels have “ns”. Pick one style. It doesn’t make the results in panel A better than the others because there’s a P value somewhat closer to p<0.05. Either report all the NS p values or none.

Response (6): We have updated the figures to report “ns” for all p values >0.05. We thank the reviewer for this catch.

(7) Line 113 refers to supplemental methods but I don’t see any supplemental methods

Response (7): We thank the reviewer for catching something we meant to remove and missed. We have removed this reference to supplemental methods in the text.

(8) More description of the ciliary beat frequency measurements is needed to determine if the authors are appropriately powered to measure CBF. What is the frame rate? Are they at or greater than Nyquist sampling rate? Is this single line scanning (they say it was done on a Zeiss LSM880)? More description of this is needed here. Just want to make sure whenever someone reports no change in CBF that they are imaging fast enough to see the changes if there are changes.

Response (8): We have added the following additional information to the CBF methods:

“Line scanning mode was used to acquire images every 1.02 milliseconds (~980 Hz). A 40x LD C-Apochromat 40x/1.1 W Korr M27 objective (Carl Zeiss) was used to obtain images. Within a single image, line scans often captured several ciliated cells… Each line scan was averaged into one value, and five line-scans were performed per donor.”

To the appropriate figure legend (Fig 2C), we have specified: “Each data point represents the average data for a single donor.”

---

## [Decision Letter · Decision Letter 1]

25 May 2022

Vitamin D-mediated effects on airway innate immunity in vitro

PONE-D-22-05766R1

Dear Dr. Comellas,

We’re pleased to inform you that your manuscript has been judged scientifically suitable for publication and will be formally accepted for publication once it meets all outstanding technical requirements.

Kind regards,

Y. Peter Di, Ph.D.

Academic Editor

PLOS ONE

Additional Editor Comments (optional):

Reviewers' comments:

Reviewer's Responses to Questions

**Comments to the Author**

1. If the authors have adequately addressed your comments raised in a previous round of review and you feel that this manuscript is now acceptable for publication, you may indicate that here to bypass the “Comments to the Author” section, enter your conflict of interest statement in the “Confidential to Editor” section, and submit your "Accept" recommendation.

Reviewer #1: All comments have been addressed

Reviewer #2: All comments have been addressed

2. Is the manuscript technically sound, and do the data support the conclusions?

Reviewer #1: Yes

Reviewer #2: Yes

3. Has the statistical analysis been performed appropriately and rigorously? 

Reviewer #1: I Don't Know

Reviewer #2: Yes

4. Have the authors made all data underlying the findings in their manuscript fully available?

Reviewer #1: Yes

Reviewer #2: Yes

5. Is the manuscript presented in an intelligible fashion and written in standard English?

Reviewer #1: Yes

Reviewer #2: Yes

6. Review Comments to the Author

Reviewer #1: (No Response)

Reviewer #2: Please use the space provided to explain your answers to the questions above. You may also include additional comments for the author, including concerns about dual publication, research ethics, or publication ethics. (Please upload your review as an attachment if it exceeds 20,000 characters) (Limit 100 to 20000 Characters)

All comments have been satisfactorily addressed.

7. PLOS authors have the option to publish the peer review history of their article (what does this mean?). If published, this will include your full peer review and any attached files.

Reviewer #1: No

Reviewer #2: No

---

## [Editor Report · Acceptance letter]

30 May 2022

PONE-D-22-05766R1 

Vitamin D-mediated effects on airway innate immunity *in vitro*

Dear Dr. Comellas:

I'm pleased to inform you that your manuscript has been deemed suitable for publication in PLOS ONE. Congratulations! Your manuscript is now with our production department. 

Kind regards, 

on behalf of

Dr. Y. Peter Di 

Academic Editor

PLOS ONE